# From Chilblains (Pernio) to Coeliac Disease—Should We Still Consider It Random?

**DOI:** 10.3390/children9121972

**Published:** 2022-12-15

**Authors:** Mario Mašić, Ana Močić Pavić, Alenka Gagro, Ana Balažin Vučetić, Suzana Ožanić Bulić, Zrinjka Mišak

**Affiliations:** 1Referral Centre for Paediatric Gastroenterology and Nutrition, Children’s Hospital Zagreb, 10000 Zagreb, Croatia; 2Department of Pulmonology, Allergology, Immunology and Rheumatology, Children’s Hospital Zagreb, 10000 Zagreb, Croatia; 3School of Medicine, University of Zagreb, 10000 Zagreb, Croatia; 4Health Centre Zagreb–Center, 10000 Zagreb, Croatia; 5Department of Dermatology, Children’s Hospital Zagreb, 10000 Zagreb, Croatia

**Keywords:** coeliac disease, pernio, gluten-free diet, dermatitis, vasculitis.

## Abstract

Coeliac disease (CD) is a gluten-triggered, immune-mediated inflammatory disease occurring in genetically predisposed individuals, causing a variety of gastrointestinal and extraintestinal symptoms. The most common cutaneous association of CD is dermatitis herpetiformis, although recent reports have sought to link CD with other dermatological and autoimmune diseases. Chilblain, also called pernio, is usually a benign, superficial and localized inflammatory skin disorder that results from a maladaptive vascular response to non-freezing cold. We present a patient with pernio (chilblains) and newly diagnosed CD, with a significant intestinal lesion–total villous atrophy, as there are only two known cases of this feature associated with CD published in the literature. In the workup of chilblains (pernio) in children, an active case finding for coeliac disease should be conducted with coeliac-specific serology testing.

## 1. Introduction

Coeliac disease (CD) is a gluten-triggered, immune-mediated inflammatory disease occurring in genetically predisposed individuals, causing a variety of gastrointestinal and extraintestinal symptoms [1,2]. The most common cutaneous association of CD is dermatitis herpetiformis, although recent reports are trying to link CD with other dermatological and autoimmune diseases [3,4]. We present a case of a patient with chilblains (pernio), which are associated with CD, as there are only two known cases of this feature associated with CD published in the literature.

## 2. Case Presentation

Our patient is a nine-year-old female who initially presented to a rheumatologist and dermatologist due to complaints of feeling coldness in her legs and fingers, with inflammatory lesions of the skin on the distal parts of the toes and fingers (Figure A1 and Figure A2). She did not have any other symptoms; her physical status was unremarkable with no abdominal distension or pain, no other skin lesions, no mouth ulcerations or any other symptoms suggestive of gastrointestinal disease. The patient’s anthropometric measurement at the time of the first exam were as follows: weight of 23.2 kg (−0.89 Z-score), height of 134 cm (0.73 Z-score), and body mass index (BMI) of 12.9 kg/m^2^ (−2.79 Z-score). The patient was a term baby, born from an uncomplicated pregnancy and is the single child in the family. She had vesicoureteral reflux followed by pediatric nephrologist, but was otherwise healthy. Her mother had thrombophilia (PAI I 5G/4G heterozygosity, MTHFR heterozygosity C/T), and father had Barrett’s oesophagus. 

After the examination of the rheumatologist and dermatologist, a diagnosis of pernio (chilblains) was proposed. Chilblains, or pernio, as the Mayo Clinic diagnostic criteria suggest, can be diagnosed if the patient has one major and at least one minor criterion. Our patient had fulfilled major criterion (localized erythema and swelling involving acral sites and persistent for >24 h), and one minor criteria (onset and/or worsening in cooler months, between November and March). A skin biopsy was not conducted on our patient, as the Mayo Clinic criteria for the diagnosis were met by other means. [5] An extensive laboratory workup was performed, including complete blood cell count, thrombocyte count, coagulation parameters, anti-nuclear antibodies (ANA), extractable nuclear antigen (ENA) and anti-neutrophil cytoplasmic antibodies (ANCA). The complement levels and activity (C3, C4, CH50), lupus anticoagulant (LAC), cryoglobulins, anticardiolipin antibodies, anti-beta2 glycoprotein (GPI) immunoglobulin M (IgM) and IgG antibodies were measured. The workup also included a color Doppler ultrasound of the lower limb extremities to exclude thrombosis and thrombophilia screening (FVIII, protein C, protein S, LAC, factor V Leiden and factor II mutation, MTHFR polymorphism) due to familial risk factors. All of the results were normal, with the exception of the ANA antibodies, which gave positive granular fluorescence (1:640) with positive fiber fluorescence in the cytoplasm. Due to the positive fibers in the cytoplasm, specific confirmatory tests for smooth muscle antibodies were performed (anti-SMA). Positive specific antibodies have been detected on a specific substrate to actin in titre 1:160 (anti-SMA reference interval: negative < 1:40). The screening for thrombophilia revealed PAI-1 4G/4G mutated homozygosity, and MTHFR AC heterozygosity. As a part of the diagnostic workup, among other tests, a screening for coeliac disease was performed with coeliac-specific serology tests, with the following results: normal total immunoglobulin A (IgA) levels of 2.38 g/L and tissue transglutaminase antibodies (tTG-IgA) of 38 U/mL (positive > 10 U/mL).

The patient was subsequently referred to our gastroenterology department. An additional laboratory workup was recommended with repeated anti-tTG, which was four times the upper limit of normal and endomysial antibodies (EMA), which were also positive, based on which an upper endoscopy was indicated, according to current guidelines (1). The patient underwent an endoscopy with small bowel biopsies, and the histology was consistent with the diagnosis of CD, Marsh stage IIIC—total villous atrophy (as shown in Figure A3). Finally, four months after the lesions first presented in the patient, based on the aforementioned findings, coeliac disease was diagnosed, and a strict gluten-free diet was introduced. As recommended by ESPGHAN, the patient was seen by a clinical nutritionist, and the supplementation of vitamin D and zinc was initiated (1). At three and seven months of follow-up, the coeliac-specific serology was declining, and the patient’s skin lesions improved. The coeliac-specific serology completely normalized after seven months of follow-up, with the levels of anti-tTG returning to normal values (<7 U/mL).

During the winter months and eight months following the CD diagnosis, the skin lesions recurred, presenting with slightly livid to erythematous patches on the dorsal aspect of her toes and mild erythema with dryness localized symmetrically to her fingertips (less intense than on initial presentation), all in accordance with a pernio (chilblains) diagnosis. The patient and her parents were asked about the possibility of accidental gluten exposure, which could exacerbate the symptoms, but the parents rejected the option as they have cleaned the house of gluten. The decision of the rheumatologist was to repeat the immunological workup, including ANA + ENA, ANCA, C3, C4, CH50, anti-beta2-GPI IgM and IgG, and anti-cardiolipin IgM and IgG antibodies. Of the aforementioned workup panel, antinuclear antibodies persisted with positive granular fluorescence (1:640), with positive fiber fluorescence in the cytoplasm. Due to the positive fibers in the cytoplasm, specific confirmatory tests were performed, and positive specific antibodies were detected on a specific substrate to actin in titre 1:160 (anti-SMA reference interval: negative < 1:40). Panel to systemic sclerosis was performed, and it returned negative. The total complement haemolytic activity was normal (95%), and the anti-cardiolipin IgM and IgG antibodies were negative. In the follow-up, approximately nine months after the initial diagnosis, the patient’s general health was unremarkable, with no symptoms present; she had gained in height (137 cm (+3 cm from the diagnosis), 0.43 Z-score), weight (24.4 kg (+1.2 kg from the diagnosis), −1.16 Z-score) and BMI of 13 kg/m^2^ (−2.76 Z-score). In the following winter months, 18 months from the diagnosis of CD, there was no reoccurrence of pernio (chilblains) symptoms (erythematous lesions and livid macules) on the fingers and toes, as shown in Figure A4.

## 3. Discussion

We presented a patient with pernio (chilblains) and newly diagnosed coeliac disease (CD), with a significant intestinal lesion–total villous atrophy. During the last few decades, physicians have been more prone to diagnosing CD in a variety of patients, as they are becoming more aware of the various possible extraintestinal symptoms, such as dermatitis herpetiformis, stunted growth, delayed puberty, amenorrhea, recurrent aphthous stomatitis, iron-deficiency anemia, irritability, arthritis, chronic fatigue, headaches, epilepsy, cardiomyopathy, and alopecia [1,2]. There are also reports of an evident shift from gastrointestinal to extraintestinal symptoms prevalent in children and adults, as some researchers suggest that up to 43% of symptoms in pediatric CD are non-intestinal [6,7]. With the advancement and availability of coeliac-specific serology tests, clinicians are trying to avoid delays in the diagnosis of CD, as it is known that untreated CD leads to greater morbidity and complications later in life [1]. Researchers are trying to classify CD-associated skin diseases, to those diseases with a proven CD association, such as dermatitis herpetiformis, and to those which are only secondary due to nutritional deficiencies of vitamins (A, D, B12, folic acid) and other micronutrient deficits (zinc, iron) [3,4]. 

Louis Duhring initially described dermatitis herpetiformis in 1983, and it is the most common skin disease associated with CD [8]. The clinical presentation is variable as the lesions can show polymorphism consisting of erythema, papules and small vesicles coalescing, into larger plaques often with an excoriated, crusted surface. Lesions are grouped symmetrically on the extensor surfaces of the arms and leg, particularly elbows, knees, as well as posterior neck, but the rash is often misdiagnosed as eczema, in particular for its itchiness. The age of onset is variable, and it can often be underdiagnosed [9]. Encouraged by this dermatological feature of CD, researchers are trying to correlate other dermatological and autoimmune diseases with CD, such as psoriasis, alopecia areata, chronic urticaria, cutaneous vasculitis, hereditary angioneurotic oedema and others, although without much success [4,10]. In a study by Dev et al., conducted on 300 coeliac disease patients, the most common skin manifestation of CD was dermatitis herpetiformis (16%), and the second was psoriasis (13.8%) [11]. To the best of our knowledge, there are only two published reports of chilblains associated with CD [12,13].

Chilblains, also called pernio, are usually benign, superficial and localized inflammatory skin disorder that result from a maladaptive vascular response to non-freezing cold. The prevalence is reported to be 0.9 to 1.7 per 1000, mostly affecting women and young to middle-aged adults. Although well described, childhood pernio seems to be uncommon and possibly under-diagnosed [14]. The aetiology of perniosis is unclear. The proposed pathogenesis of this condition is defective vasodilatation induced by cold, resulting in localized inflammation in response to hypoxic damage to the tissue. It is suggested that low body mass index could be a predisposing factor [14,15,16,17]. Unfortunately, there are no large pediatric epidemiologic studies. In one study by Takci et al., conducted on 51 patients, chilblain was primarily an idiopathic condition (in 86% of cases), and in the rest of the cases it was secondary due to hepatitis or connective tissue disorder. In the same study, the clinical features of secondary perniosis were associated with hypogammaglobulinaemia, photosensitivity, older age and persistence of symptoms after the cold periods [18]. Other studies have also described perniosis as a symptom of systemic lupus erythematosus [19]. Our patient also presented with chilblains, which persisted after the cold periods, with no other leading symptoms. The chilblains were localized on the toes and fingers, with distinctive erythematosus patches, as seen in Figure A1 and Figure A2.

In accordance with the literature’s recommendations, we performed an extensive workup and screening for possible associated and underlying autoimmune diseases [14]. A risk for thrombophilia was found and positive specific antibodies to actin in titre 1:160. The haematologist was consulted and proposed a further follow-up due to the positive risk of thrombophilia. Pertaining to positive anti-actin antibodies, the rest of the workup was normal, including normal liver enzymes, normal immunoglobulins levels, and no signs of autoimmune hepatitis. As shown in the literature, anti-actin antibodies can be indicative of severe intestinal damage in CD. Although the sensitivity of anti-actin antibodies in CD is low, they can be used to confirm the suspicion of CD in difficult cases [20]. Systemic erythematosus lupus and systemic sclerosis were excluded. In addition, although our patient did not have any other symptoms, based on the literature search and two case reports on the possible association with coeliac disease, we performed the screening. Finally, CD was diagnosed according to the ESPGHAN criteria (positive coeliac-specific serology—positive anti-tTG and EMA antibodies and Marsh IIIC–total villous atrophy on small bowel biopsy). Six months after initiating a strict gluten-free diet, the coeliac-specific serology was negative with anti-tTG 4.6 U/mL (negative < 10 U/mL), which is regarded as a good laboratory response.

The question of whether pernio and coeliac disease are pathogenically related remains open. The patient described by St. Clair et al. had significant weight loss, which might have been the predisposing factor for chilblains [12]. In contrast, the case report by Lemieux et al. and our case did not have this predisposing factor [13]. The mechanism of peripheral vasoconstriction and cutaneous vasoreactivity due to weight loss is well described in patients with anorexia nervosa. Those patients experience weight loss, which leads them to a hypothermic state, resulting in acrocyanosis (bluish, painless discoloration of fingers and toes) or lesions similar to chilblains. With appropriate weight gain, in those patients, the symptoms perish. [21,22]. It is debatable whether a similar mechanism could explain the symptoms in our patient, because significant weight loss was not observed, and the chilblains were resolved before significant weight gain occurred. The malabsorption of nutrients in CD is presumed to primarily be the result of intestinal atrophy, the loss of a functional bowel barrier and other immunological changes. Furthermore, the presence of skin lesions in patients with CD could also be secondary, due to the malabsorption of vitamins (A, D, B12, folic acid) and other oligoelemental deficit (zinc, iron). Our patient had vitamin D levels of 53.5 nmol/L (normal > 75 nmol/L) and had no iron deficit, while other elements were not measured. The vitamin and mineral deficit and associated dermatological sequelae are known in different diseases, such as eating disorders, multiple nutritive allergies, chronic intestinal inflammatory disease (coeliac disease, IBD), acrodermatitis enteropathica and cystic fibrosis [23,24,25]. Vitamin B12 deficiency has been reported in 5–19%, folic acid in 14–31%, vitamin D 0–70%, iron in 12–82%, and zinc in more than 50% of untreated CD pediatric patients [4,26,27]. Overall, nutritional deficits are highly prevalent in untreated CD patients, for which we emphasize the importance of the role of clinical nutritionists in further follow-ups for this group of patients.

Although chilblains can resolve spontaneously, the treatment options for those that do not resolve are limited and not always successful. The first step is avoiding cold and keeping extremities warm and dry. If these measures fail to improve the symptoms, topical corticosteroids and calcium channel blockers (nifedipine) may be tried [14]. On the other hand, the treatment of coeliac disease is a lifelong, strict gluten-free diet. As in two other cases [12,13], the chilblains in our patient improved with the implementation of a gluten-free diet, indicating a possible correlation and highlighting the question of whether chilblains represent a newly recognized, extraintestinal, dermatologic manifestation of coeliac disease. However, keeping in mind the possibility of the spontaneous resolution of chilblains, it is difficult to draw a definite conclusion. Further studies are needed to help us better understand the possible association between these two entities. Finally, we emphasize the importance of recognizing this feature, as our patient could not have otherwise been diagnosed with CD as she lacked any other typical symptoms.

## 4. Conclusions

Coeliac disease (CD) can present with a variety of intestinal and extraintestinal symptoms, and the most common cutaneous feature of CD is dermatitis herpetiformis. Previous reports have suggested that most of the dermatological associations are likely secondary, due to vitamin and other nutrient deficits; however, in the last few decades, the need for further investigation into the association of different dermatological and immunological features with CD has emerged. In the workup of chilblains (pernio) in children, an active case finding for coeliac disease should be conducted with coeliac-specific serology testing.

## Data Availability

Not applicable

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
