# Peer review of "From Chilblains (Pernio) to Coeliac Disease—Should We Still Consider It Random?"

_children, 2022, doi:10.3390/children9121972_

Round 1

Reviewer 1 Report

The authors present an interesting case report with chilblains that improved with gluten-free diet, indicating a possible correlation and opening question whether chilblains represent a newly recognized extraintestinal, dermatologic manifestation of coeliac disease.

MAJOR COMMENTS:  

1)      The authors slhoud include if the patient had any episode associated with accidental gluten exposure. Since the introduction of this diet, the patient dodn’t demonstrate any symptoms of pernio in the following at least four winters despite exposition to cold temperature. One episode of symptoms should be correlated with an accidental exposure to gluten by the patient

2)      The authors must include in the work the physical examination found with a weight (percentile), a height percentile), and a calculated body mass index.  

3)      The authors must include a new figure in which show the complete resolution of the lesions of chilblains  few months after the gluten-free diet was introduced.

4)      The authors must include a new figure correspondoing a biopsy specimen of the duodenum revealed villous atrophy, crypt hyperplasia, increased intraepithelial lymphocytes, and lymphoplasmacytic expansion of the lamina propria, consistent with celiac disease

5)      The authors must include a new figure with the description of Skin biopsy specimen of chilblains with the presence of infiltrate with some neutrophils, marked papillary edema, and dyskeratotic cells.for the diagnosis of chilblains

6)      The authors must include the period of time to present the lesions on the distal surfaces of her fingers and toes in this case report

7) In the discussion chapter, the authors must consider that similar thermoregulatory defects have also been described in malnourished patients who experience weight loss, it is possible that weight loss from celiac disease led to a similar malnourished, hypothermic, and vasoreactive state, resulting in an increased susceptibility to chilblains. Notably, maintenance of a gluten-free diet led to weight gain and resolution of the chilblains.

Author Response

Dear Reviewer,

We are sending our comments and revised manuscript entitled “From chilblains (pernio) to coeliac disease – should we still consider it random?” for possible publication in the journal Children. We would like to thank you for your valuable comments, which are all addressed in detail below. We hope that you will find the revised manuscript suitable for publication in your esteemed journal.

Please see the attachment below.

Kind regards,

Authors

Reviewer 2 Report

The diagnosis does not indicate whether genetic testing of the child and his parents was carried out. I would like to clarify whether genetic typing for the presence of alleles was carried out HLA-DQ2/DQ8.

Can the described case of pernio be explained by hereditary thrombophilia, which is described in the mother. Has research been done in this direction?

I would also like to see the answer to the gluten-free diet. was efficiency control carried out by the titer of specific antibodies or in some other way?

Author Response

(The authors gave the same response as above.)
